# A Highly Sensitive, Low Creep Hydrogel Sensor for Plant Growth Monitoring

**DOI:** 10.3390/s24196197

**Published:** 2024-09-25

**Authors:** Haoyan Xu, Guangyao Zhang, Wensheng Wang, Chenrui Sun, Hanyu Wang, Han Wu, Zhuangzhi Sun

**Affiliations:** Province Key Laboratory of Forestry Intelligent Equipment Engineering, College of Mechanical and Electrical Engineering, Northeast Forestry University, Harbin 150000, China; xhaoyan58@gmail.com (H.X.); 18845796925@163.com (G.Z.); wangwensheng@nefu.edu.cn (W.W.); 18724669768@163.com (C.S.); keepwhy@nefu.edu.cn (H.W.)

**Keywords:** dual−network structure, low creep, high sensitivity, plant growth monitoring

## Abstract

Ion−conducting hydrogels show significant potential in plant growth monitoring. Nevertheless, traditional ionic hydrogel sensors experience substantial internal creep and inadequate sensitivity, hindering precise plant growth monitoring. In this study, we developed a flexible hydrogel sensor composed of polyvinyl alcohol and acrylamide. The hydrogel sensor exhibits low creep and high sensitivity. Polyvinyl alcohol, acrylamide, and glycerol are crosslinked to create a robust interpenetrating double network structure. The strong interactions, such as van der Waals forces, between the networks minimize hydrogel creep under external stress, reducing the drift ratio by 50% and the drift rate by more than 60%. Additionally, sodium chloride and AgNWs enrich the hydrogel with conductive ions and pathways, enhancing the sensor’s conductivity and demonstrating excellent response time (0.4 s) and recovery time (0.3 s). When used as a sensor for plant growth monitoring, the sensor exhibits sensitivity to small strains and stability for long−term monitoring. This sensor establishes a foundation for developing plant health monitoring systems utilizing renewable biomass materials.

## 1. Introduction

The sensors are increasingly used in plant growth monitoring and smart agriculture. Over time, substantial progress in rigid electronic components and materials has resulted in inherently inflexible sensor systems, which have standardized their operational principles and environmental adaptability. In previous sensor applications, the primary requirements were stability, cost −effectiveness, and durability. Rigid sensors effectively met these requirements and generally fulfilled these criteria. Despite the maturity of these sensor technologies, the intrinsic form and material limitations of rigid sensors lead to drawbacks such as bulkiness and fragility. These limitations hinder their applicability in flexible human−machine interactions and portable wearable smart devices [1,2,3,4,5]. In recent years, driven by continuous technological advancements, flexible sensors have garnered widespread attention for their excellent biocompatibility and ductility [6,7,8,9,10,11,12]. Researchers have sought to integrate flexible sensors with plants to address the limitations of traditional rigid sensors [13,14,15,16,17,18,19,20,21,22]. Gels are a class of soft materials with three−dimensional (3D) polymer networks capable of sustaining various dispersion media. Among them, hydrogels with water as the dispersion medium have been widely used as flexible sensors due to their unique advantages, such as good transparency, high stretchability, low hysteresis, and outstanding biocompatibility [23,24]. However, traditional hydrogel sensors often produce signal drift under long−term stress, leading to decreased monitoring accuracy and unstable data. Signal drift arises from the creep of polymerized chains within the hydrogel under stress, causing irreversible alterations in sensor resistance. Therefore, the development of a highly sensitive sensor with low creep characteristics is essential for accurate and stable plant growth monitoring.

Polyvinyl alcohol (PVA) has been widely investigated for flexible sensor research due to its high chemical stability, good biocompatibility, low toxicity, and low cost [25,26,27,28,29,30,31]. Generally, single−network hydrogels formed by PVA are unstable due to their network structure. Consequently, the viscoelastic creep of the hydrogel under external stress can induce signal drift and lead to measurement inaccuracies over extended periods, rendering precise monitoring impractical. It has been demonstrated that the formation of dual−network composite hydrogels by doping is an excellent strategy [32,33,34,35,36,37]. Zhu et al. developed a PVA hydrogel using a one−pot freeze−thawing method, cross−linked through hydrogen bonding and interactions between hydroxyl groups. They introduced iron ions to enhance electrical conductivity, resulting in satisfactory high sensitivity and durability [38]. This hydrogel comprises a polymer network with two cross−linking mechanisms. The primary cross−linking sites provide stable mechanical strength, while the weaker secondary cross−linking sites serve as an energy dissipation mechanism, imparting toughness to the hydrogel [39,40,41]. The robust dual−network structure enhances the cross−linking density and reduces hydrogel creep by suppressing network structure fluctuations. In addition, when cross−linking occurs in acrylamide, its cross−linked network structure contains many amide bonds, making it easy to form hydrogen bonds. At the same time, the network space between the polymer chains provides sites for various functional molecules, which makes it easy to functionalize. However, its application is limited due to its low strength. Introducing another matrix material to prepare dual network hydrogels is an effective method to improve the mechanical properties. Therefore, by crosslinking polyvinyl alcohol and acrylamide in a low−temperature environment, hydrogels with a strong crosslinked skeleton are obtained. In addition, internal ions are an important factor affecting electrical conductivity, which is achieved by adding conductive ions to the network and constructing conductive pathways to control the rapid migration of ions.

In this work, a low creep polyelectrolytic hydrogel sensor (SNaPVA−Sensor) was proposed, which significantly reduces the signal drift and possesses excellent sensing performance. The SNaPVA−Sensor utilizes a dual−network structure formed by polyvinyl alcohol (PVA), acrylamide (AAm), and malonyl alcohol, reducing the hydrogel’s creep under stress. AgNWs and sodium chloride (NaCl) immersion treatments were also employed to enhance the electrochemical properties of the SNaPVA−Sensor and improve its sensing sensitivity. The results showed that the conductivity of the SNaPVA−Sensor was enhanced, while signal drift was suppressed to varying degrees. Finally, the prepared hydrogels successfully monitored different stem widths and plant growth processes, demonstrating stable sensing performance. This work provides new optimization strategies for plant growth monitoring, promoting intelligent forestry development.

## 2. Materials and Methods

### 2.1. Materials

Polyvinyl alcohol (PVA, molecular weight 1750 ± 50) was purchased from Sinopharm Chemical Reagent Co., Ltd. (Shanghai, China); acrylamide (AAm, molecular weight 71.08) was purchased from Tianjin Reagent Factory (Tianjin, China), sodium chloride (NaCl, molecular weight 58.5) was purchased from China Yongchang Reagent Co (Harbin, China).; silver nanowires (AgNWs) were purchased from Nanjing Suzhan Intelligent Technology Co (Nanjing, China).; N,N′−Methylenebisacrylamide (molecular weight 154.17) was purchased from the Tianjin Guangfu Institute of Fine Chemical Industry(Tianjin, China); N,N,N,N−Tetramethyl Ethylenediamine (molecular weight 116.21) was purchased from Tianjin Damao Chemical Reagent Factory (Tianjin, China); ammonium persulfate (molecular weight 228.201) was purchased from Fuchen (Tianjin) Chemical Reagent Co. (Tianjin, China); and glycerine (molecular weight 92.09) was purchased from Tianjin Fuyu Fine Chemical Co (Tianjin, China).

### 2.2. Material Characterization

The functional groups of the test samples were analyzed in the spectral range of 4000–500 nm using a Nicolet iS50 Fourier transform infrared spectrometer (FT−IR, Thermo Fisher, Waltham, MA, USA). For SNaPVA−Sensor, OH stretching vibration usually appears in the wavenumber range of 3000–3600 cm^−1^, indicating the presence of hydroxyl (OH) functional groups in the material. Changes in this vibration frequency can reflect the formation or destruction of hydrogen bonds in the material. The C=O absorption peak generally appears between 1600–1750 cm^−1^, indicating the presence of carbonyl (C=O) in the material. The appearance of this absorption peak usually indicates that the material contains functional groups such as aldehydes, ketones, carboxylic acids, or esters [42]. The test samples were analyzed using an X, Pert3 Powedr X-ray diffractometer (XRD). The crystallinity is higher when the peak 2Ɵ is closer to 20°, indicating stronger mechanical properties and higher stability [43]. The tensile performance of the SNaPVA−Sensor was determined by fixing one end and pulling horizontally. The modulus and tensile strength are measured by the SNaPVA−Sensor and were measured using a digital push—pull tester (HP−500, Zhejiang, China).

### 2.3. Electrical Response Characterization

The sensors were used to establish the electrochemical properties of the experimental samples by cyclic voltammetry (CV) and alternating impedance method (EIS) using a Corrtest CS350H multichannel chemical test station (Kesite, Zhongshan, Guangdong, China). The symmetry of CV can evaluate the reversibility of the reaction. EIS is used to measure the material’s conductivity and charge transfer efficiency and analyze the material’s impedance behavior at different frequencies [44]. The sensing signals of the sensors were obtained with a Keithley 6514 electrostatic meter (Keithley, Beaverton, OR, USA). The electrometer can reflect the resistance change of the SNaPVA−Sensor, and through the resistance change, we can know the ability of the SNaPVA−Sensor to monitor plant growth [45].

### 2.4. Preparation of PVA−Sensor

As shown in Figure 1a, the precursor solution for the PVA−Sensor was prepared by adding 2 g of polyvinyl alcohol (PVA) and 1 g of acrylamide (AAm) to 20 milliliters of deionized water. This mixture was thoroughly stirred at a temperature of 90 °C to ensure complete dissolution and uniformity. During this process, sodium chloride (NaCl) and silver nanowires (AgNWs) were incorporated into the solution to create the precursor solutions for NaPVA−Sensor and SNaPVA−Sensor, respectively.

Following the initial preparation, specific amounts of cross−linking agents and initiators were added to the mixture. These included 0.05 g N,N′−methylene bisacrylamide (MBA), 0.1 mL N,N,N′,N′−tetramethylethylenediamine (TEMED), 0.006 g ammonium persulfate (APS), and 2 mL glycerol. These components were added at room temperature, and the mixture was stirred until it achieved a viscous liquid consistency.

To finalize the preparation, the viscous solution was subjected to a freeze−thaw treatment. This involved placing the solution in an environment maintained at −20 °C. After the solution was thoroughly frozen, it was then thawed, resulting in the formation of the PVA−Sensor. This freeze−thaw process is critical as it helps in enhancing the mechanical properties and stability of the final hydrogel sensor. During the freezing process, the PVA molecular chains form crystalline regions, which are cross−linked by hydrogen bonds to form the three−dimensional structure of the hydrogel. When the temperature rises to room temperature, these crystalline regions remain stable and will not be destroyed by temperature changes [46,47]. The PVA and AAm interact to form a physical cross−linking network, while adding glycerol promotes dynamic hydrogen bonding, establishing a multi−cross−linking network within the polymer matrix. The dynamic hydrogen bonding enhances the mechanical integrity of the hydrogel. The dual−network structure imparts low creep to the hydrogel due to its high crosslink density, which inhibits polymer chain fluctuations. Additionally, the increased chain density improves entropic elasticity, collectively reducing hydrogel creep. Furthermore, NaCl is used as a kinetic ion to enhance sensing performance, and adding AgNWs significantly improves the electrical conductivity of the SNaPVA−Sensor (Figure 1b).

### 2.5. Electrochemical Testing

Electrochemical tests were conducted using a two−electrode method. The SNaPVA−Sensor was immersed in 1 mol/L LiCl solution, with nickel foam attached to the exterior as the conductive electrode. AC impedance curves of the PVA−Sensor, NaPVA−Sensor, and SNaPVA−Sensor were tested at different scan rates. Subsequently, EIS curves for all three sensors were measured in the frequency range of 10^5^–0.01 Hz. Afterwards, the specific capacitance of different PA−iEAP was calculated using the following formula [48]:(1)$$C=\frac{1}{2\cdot s\cdot r\cdot \Delta V}\int_{V_{0}}^{V_{0}+\Delta V}IdV$$
where S is the surface area of the electrode, r is the voltage scan rate, ΔV is the potential drop throughout the cycle, and $V_{0}$ is the minimum voltage during the cycle.

### 2.6. Sensing Performance Testing

The drift ratio and drift rate of SNaPVA−Sensor are calculated by measuring the resistance change value through Keithley 6514. The specific formula is as follows:$$Drift\;ratio=\frac{\Delta R}{R_{0}}$$
$$Drift\;rate=\frac{d\left(R/R_{0}\right)}{dt}$$

The sensor sensitivity of the SNaPVA−Sensor is obtained through the Keithley 6514 test. The sensor is connected to the electrometer through a platinum electrode. When external pressure acts on the sensor, the pressure signal is converted into a resistance signal and transmitted to the electrometer, and finally the internal resistance change of the sensor is obtained.

## 3. Results

### 3.1. Preparation and Function of SNaPVA−Sensor

The hydrogel sensor wraps tightly around the plant’s surface. When the plant grows, the hydrogel experiences outward stress. Local pressure compresses the hydrogel’s double−network structure, changing the rate at which internal ions move. This change in ion mobility affects the ion concentration within the network, thereby changing its internal resistance. This mechanism enables long−term, stable plant growth monitoring (Note S1). Traditional hydrogel sensors have sparse networks that allow free ion movement. When subjected to steady pressure, the movement of internal ions can still affect resistance, causing signal drift. This drift undermines the stability of the sensor during long−term monitoring. Therefore, we prepared a low −creep hydrogel sensor to minimize signal drift as much as possible (Figure 2a). In addition, due to the strong cross−linking between polymer chains, SNaPVA−Sensor maintains good tensile properties, and the stretching rate can reach 140% when one end is fixed and stretched to the other side (Appendix A). At the same time, the Young’s modulus and tensile strength can reach 118 kPa and 165 kPa respectively (Appendix A). The results show that the sensor has good stretchability and deformation response ability, and can undergo significant deformation under stress conditions [49,50].

Figure 2b shows that the SNaPVA−Sensor has stronger and narrower crystalline peaks, particularly around 2θ = 26°, indicating higher crystallinity and a more regular structure. This structure reduces the free movement of molecular chains due to high−density cross−linking and increases overall density. Thus The FT−IR (Figure 2c) showed that at 3300 cm^−1^, the SNaPVA−Sensor exhibited a weaker broad absorption peak, indicating weakened telescopic vibrations of O−H. This weakening is due to NaCl introduction, which alters the ionic environment and affects the local O−H environment. Additionally, Na^+^ weakened the −OH strength through electrostatic interactions [51]. The addition of AgNWs introduced extra physical cross−linking points that connect PVA and AAm chains within the hydrogel network [52]. These cross−linking points limit the degrees of freedom of O−H, thus attenuating the telescopic vibrations of O−H. Finally, all three groups exhibit strong absorption peaks in the range of 1400 cm^−1^ to 1600 cm^−1^, suggesting that the high cross−linking density restricts the movement of the polymer chains, allowing more chemical bonds (e.g., O−H, C−H, C=O, etc.) to be in a more stationary and ordered environment [53]. This restriction leads to more consistent vibrations of these chemical bonds and increased intensity of the absorption peaks. In summary, the SNaPVA−Sensor has a robust cross−linked network and high electrical conductivity, facilitating the sensor’s ability to accurately monitor plant growth.

### 3.2. Electrochemical Properties of SNaPVA−Sensor

Utilizing the high conductivity obtained from the crosslinked bipolymer network structure, the SNaPVA−Sensor is expected to exhibit excellent electrochemical performance. Figure 3a–c shows the cyclic voltammetry (CV) curves of the SNaPVA−Sensor at scan rates of 20 mV s^−1^, 50 mV s^−1^, 100 mV s^−1^, with the scan voltage range set from 0 to 0.3 V. The CV curves indicate that the SNaPVA−Sensor is rectangular and exhibits a high current response at different scan rates. The curve shape is stable without obvious redox peaks, demonstrating good electrochemical stability and reversibility. In contrast, the conventional PVA−Sensor and NaPVA−Sensor showed lower current responses, while the SNaPVA−Sensor exhibited higher electrochemical activity. According to Equation (1), adding NaCl and AgNWs enhanced the specific capacitance of the PVA hydrogel (Figure 3d). NaCl addition introduced many mobile ions into the hydrogel, enhancing ion transport. On the other hand, the AgNWs formed a continuous conductive network on the surface of the cross−linked network, providing more ion conduction paths and ensuring efficient ion transport between the electrodes. These combined effects result in a larger CV curve area and higher specific capacitance for the SNaPVA−Sensor.

The EIS curves of the hydrogel sensors are shown in Figure 3e. The results indicate that in the mid−frequency Warburg impedance region, the curve’s slope is close to the imaginary axis, suggesting a relatively smooth surface of the SNaPVA−Sensor. The slope in the low−frequency range represents the ion diffusion speed. With the addition of NaCl and silver nanowires, enhanced conductivity increases ion diffusion speed, thereby improving sensing accuracy. In summary, the SNaPVA−Sensor has a larger slope in the low−frequency region, lower ion diffusion impedance, and higher ion transfer efficiency. The SNaPVA−Sensor has an equivalent resistance R_e_ of 3.773 Ω, significantly lower than the other sensors, indicating that AgNWs created a conductive network and reduced resistance (Figure 3f). Meanwhile, the low charge transfer resistance R_ct_ (15.689 Ω) reduced the resistance to ionic motion, allowing sensing to proceed efficiently (Figure 3g). Conductivity (σ) is used to measure the electrical conductivity of a material and is an important indicator for evaluating the performance of a sensor. PVA itself is poorly conductive, and the high cross−linking density in the dual−network structure of the PVA−Sensor restricts the free movement of polymer chains and provides a stabilizing framework. Some ionic conductivity was enhanced by introducing NaCl, but the enhancement was limited. AgNWs have excellent electrical conductivity. In SNaPVA−Sensor, AgNWs formed a conductive network throughout the hydrogel matrix, which greatly enhanced the overall conductivity. The high electrical conductivity of AgNWs and their uniform distribution in the hydrogel ensured efficient electron transport, and the synergistic effect of the three components significantly enhanced conductivity. Finally, the water content of the three groups was tested and found to be roughly 50–60% for all three. The introduction of glycerol, which removes a portion of the water in the hydrogel using a solvent replacement strategy, ensured the durability of the hydrogel in subsequent sensing applications.

### 3.3. Sensing Performance of SNaPVA−Sensor

We use two metrics, drift ratio and drift rate (Figure 4a) [54]. The SNaPVA−Sensor initially generates resistance R_1_ in stress response, which then drifts to R_2_ over time. Drift ratio is the ratio of the amount of resistance drift (∆R) to the initial resistance value, R_1_. The drift rate is expressed as the tangent between R_2_ and R_1_, i.e., dR/dt. To characterize the drift ratio over time, we use it as a function of the initial resistance value change over time. Figure 4b shows the drift ratios of different sensors. Under the same pressure, our hydrogel sensor’s drift ratio is over 60% lower than that of conventional sensors. This is due to the dual−network structure and high crosslinking density, which significantly suppress the hydrogel creep and reduce signal drift. Meanwhile, the drift ratio of SNaPVA−Sensor is significantly lower than that of traditional hydrogel (Figure 4c). The low drift rate indicates higher signal stability and reliability for the SNaPVA−Sensor, making it suitable for long−term monitoring. Figure 4d shows the resistance change of the SNaPVA−Sensor under different micro stresses. As external pressure increases, sensor resistance rises, indicating its ability to differentiate stresses across a wider range. Figure 4e demonstrates the response and recovery times of the SNaPVA−Sensor at a specific pressure. The sensor has a response time of 0.4 s and a recovery time of 0.3 s after applying pressure. The fast response and recovery times indicate that the SNaPVA−Sensor can monitor pressure changes in real−time with excellent dynamic performance.

In Figure 4f, the resistance change of the hydrogel shows good repeatability and stability during 300 cycles of continuous stretch and release tests at the same stress. The good linear relationship between resistance change and strain can simplify the calibration process and avoid external interference, thus ensuring the accuracy and reliability of the detected signal (Figure 4g). The SNaPVA−Sensor exhibits a sensitivity to pressure of 0.0583 and a linearity of 0.973, providing a high degree of accuracy. Overall, by introducing a dual−network structure and enhanced electrical conductivity, the SNaPVA−Sensor significantly outperforms conventional hydrogel sensors in both signal drift reduction and sensing sensitivity. These excellent properties make the SNaPVA−Sensor particularly suitable for long−term, stable plant growth monitoring.

Overall, the SNaPVA−Sensor positively correlates with pressure changes, effectively detecting and responding to various pressure levels. It can accurately identify low stresses ranging from 10 to 60 g, making it highly sensitive to minor pressure variations, which is crucial for precise pressure measurement applications.

Additionally, the sensor has a high signal−to−noise ratio, indicating its reliable and stable performance. This high ratio ensures the sensor produces clear, accurate signals with minimal background noise interference. This stability and reliability are crucial for long−term monitoring, especially in plant growth applications. The SNaPVA−Sensor enables consistent and accurate data collection, essential for assessing subtle changes in plant growth over time.

### 3.4. Application of SNaPVA−Sensor

The SNaPVA−Sensor has low creep and high sensitivity, which has great potential in monitoring plant growth applications. As shown in Figure 5a, the SNaPVA−Sensor is wrapped around the stem. As the plant grows, it expands and presses the hydrogel outward, causing the resistance of the SNaPVA−Sensor to gradually increase, thus enabling plant growth monitoring. Figure 5b shows a practical application scenario of the SNaPVA−Sensor in plant stem monitoring, where the growth of the stem expands its circumference and exerts tensile stress on the SNaPVA−Sensor, resulting in an increase in ∆R/R_0_. Figure 5c,d show the SNaPVA−Sensor’s response to stem growth over short and long intervals. The results fluctuate due to the discontinuous nature of plant growth and the gradual extension of plant cells. We manually monitored the stem growth of the plant, which increased by 0.64 mm in 4000 s. The results were slightly lagged due to issues such as inaccurate positioning during manual measurements. In contrast, the SNaPVA−Sensor offered more accurate and real−time data, surpassing traditional manual methods (Figure 5e). Figure 5f demonstrates the resistance change of the SNaPVA−Sensor during prolonged monitoring (4000 s). The results show that the sensor maintains a stable resistance change over a long period, indicating good durability and stability, making it suitable for long−term plant growth monitoring. Figure 5g demonstrates the resistance ratios under different stem widths (Group 1, Group 2, Group 3). The results show that the changes in plant stem widths across different groups can be accurately monitored by the SNaPVA−Sensor, exhibiting good recognition accuracy. The above results indicate that the SNaPVA−Sensor is an efficient and reliable sensor for plant growth monitoring, capable of real−time monitoring of plant growth.

## 4. Discussion 

A low−creep, high−sensitivity hydrogel flexible sensor, known as the SNaPVA−Sensor, has been proposed for real−time monitoring of plant growth. This innovative sensor offers significant advancements over existing technologies, particularly in sensitivity and stability. Validation tests have demonstrated that the SNaPVA−Sensor markedly improves both of these critical parameters, making it an excellent choice for precise monitoring applications. The double network structure can significantly reduce the creep of hydrogels. The main reason is that the first network layer (polyvinyl alcohol−acrylamide) provides rigid support and limits the permanent deformation of the material. The second network layer (hydrogen bonds) disperses stress, prevents stress concentration, and provides recovery ability. Therefore, the double network’s synergistic effect effectively reduces creep occurrence. Finally, the double network structure dissipates energy through the deformation of the soft network, further reducing the creep effect of the material under repeated stress [55]. The SNaPVA−Sensor effectively reduces hydrogel creep, which is a common issue in conventional hydrogel sensors. Creep can lead to gradual deformation under constant stress, affecting the accuracy and reliability of the sensor over time. By minimizing creep, the SNaPVA−Sensor maintains its structural integrity and performance consistency, ensuring long−term reliability. Furthermore, the SNaPVA−Sensor suppresses signal drift during sensing. Signal drift can occur due to environmental factors or inherent material properties, leading to inaccurate readings over prolonged periods. The ability of the SNaPVA−Sensor to mitigate this drift is crucial for reliably monitoring plant growth, which involves detecting minute deformations. By reducing signal drift, the sensor enhances monitoring accuracy, providing more reliable data for researchers and agricultural professionals.

Besides its stability, the SNaPVA−Sensor offers high sensitivity, responding precisely to different stress levels. This sensitivity detects even the smallest stress or deformation changes, crucial for applications needing detailed monitoring. This capability is particularly valuable in plant growth monitoring, where detecting slight variations can provide critical insights into plant health and development.

The practical significance of the SNaPVA−Sensor lies in its ability to deliver more precise and stable sensing signals. This leads to better assessments of plant growth, enabling more informed decisions and interventions. The enhanced performance of the SNaPVA−Sensor improves the accuracy of plant monitoring and contributes to more efficient and effective agricultural practices.

Compared to existing studies, our prepared SNaPVA−Sensor significantly improves sensitivity (Figure 6) [56,57,58,59,60]. SNaPVA−Sensor can achieve fast response within 0.4 s. Unlike conventional hydrogel sensors, the SNaPVA−Sensor features a robust dual−network structure. This structure employs a solvent replacement strategy to reduce moisture content, thereby enhancing the sensor’s service life and stability. A key advantage of the dual−network structure is its ability to increase cross−linking density, which is crucial in suppressing polymer chain fluctuations. This increased density also greatly enhances the sensor’s entropic elasticity, providing better performance under various stress conditions. The combination of reduced polymer chain fluctuations and improved entropic elasticity effectively minimizes the hydrogel’s creep, making the SNaPVA−Sensor more reliable and consistent over time. However, we still cannot achieve zero creep in hydrogel sensors, so there will still be a small amount of drift signal.

Future research may focus on achieving creep−free performance and developing multifunctional coupled sensing. Optimizing the internal ion pathways of hydrogels could achieve creep−free performance. Additionally, multifunctional coupled sensing can comprehensively assess plant physiological states by monitoring microenvironmental factors such as temperature, humidity, and various gas concentrations. This capability would facilitate timely interventions for plant growth. By coupling multiple sensing capabilities and optimizing ion channels, the SNaPVA−Sensor can further enhance its sensing accuracy and stability, resulting in improved plant monitoring outcomes.

## 5. Conclusions

In this study, we developed a low−drift, high−sensitivity dual−network hydrogel sensor (SNaPVA−Sensor) with a cross−linked PVA and AAm network, enhanced by AgNWs and NaCl. The hydrogel significantly enhanced the sensing sensitivity of the SNaPVA−Sensor by reducing creep behavior during stress through a robust dual−network structure, aided by the addition of NaCl and AgNWs. Compared to conventional hydrogels, the SNaPVA−Sensor has a lower drift rate, with a response time of 0.4 s and a recovery time of 0.3 s. It also has good longevity and stability, maintaining a stable cyclic response for 4000 s under the same stress. In addition, the SNaPVA−Sensor exhibits a tensile elongation of more than 140%. The sensor monitors plant growth in real time, promptly responding to small stresses and detecting minor strains effectively. The excellent performance of the SNaPVA−Sensor enables it to advance plant growth monitoring and promotes the development of smart agriculture and plant physiology research.

## Figures and Tables

**Figure 1 sensors-24-06197-f001:**
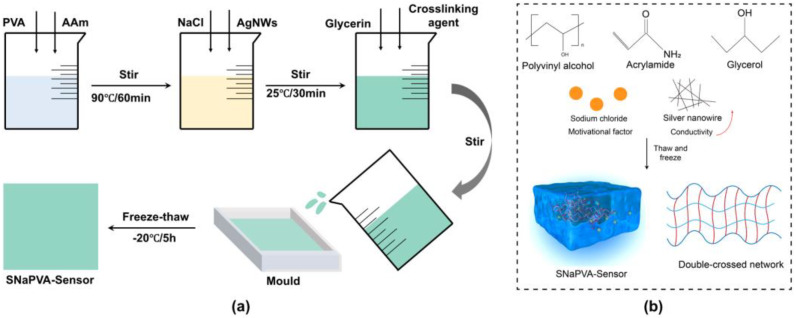
Preparation process of SNaPVA−Sensor. (**a**) Preparation process of SNaPVA−Sensor. (**b**) Cross−linking status of SNaPVA−Sensor.

**Figure 2 sensors-24-06197-f002:**
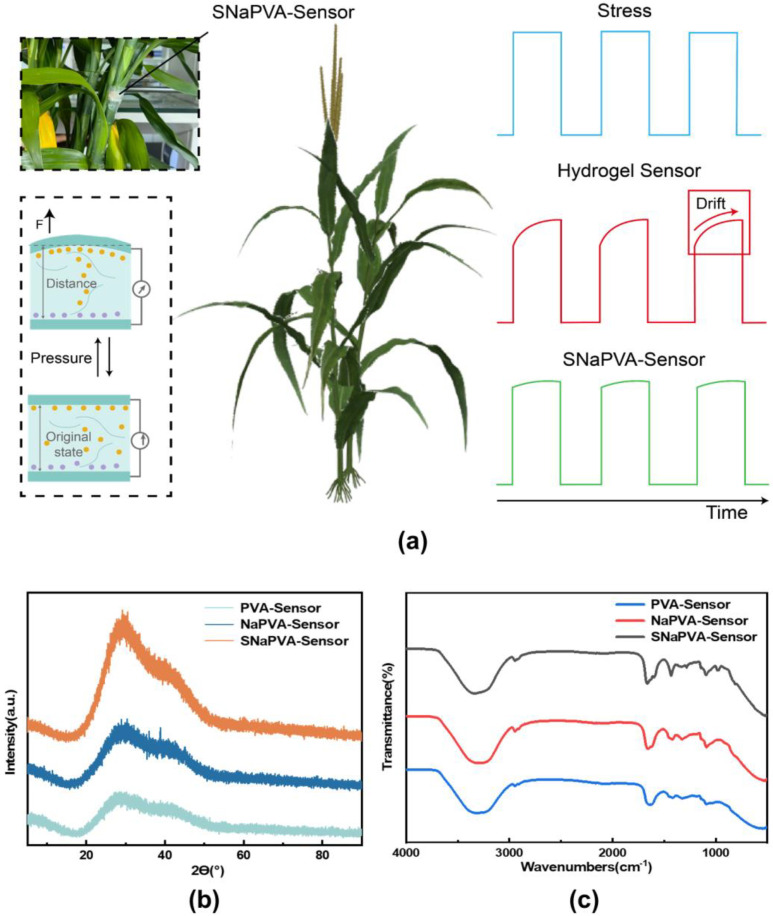
Preparation and function of SNaPVA−Sensor. (**a**) Schematic diagram of the properties and functions of the SNaPVA−Sensor. (**b**) XRD spectrum of SNaPVA−Sensor. (**c**) Infrared diffraction spectrum of SNaPVA−Sensor.

**Figure 3 sensors-24-06197-f003:**
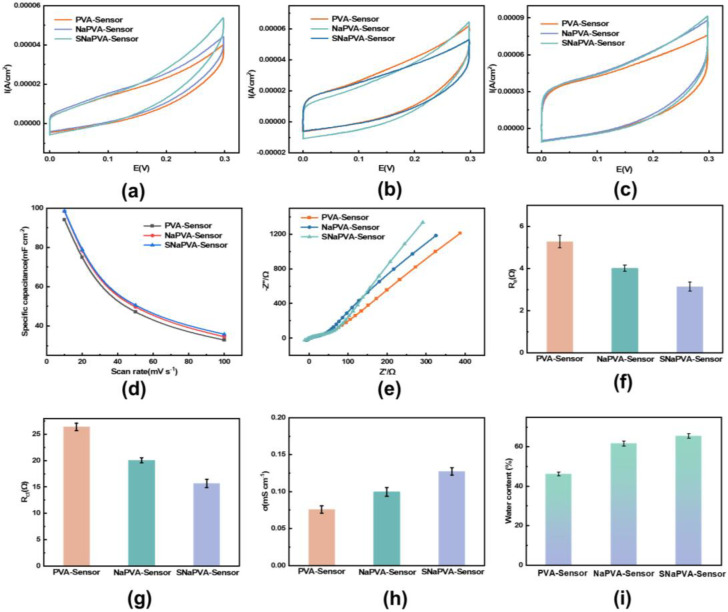
Electrochemical performance of the SNaPVA sensor. (**a**–**c**) CV curves of PVA−Sensor at 20 mV s^−1^, 50 mV s^−1^ and 100 mV s^−1^; (**d**) Capacitance of the PVA—Sensor; (**e**) EIS curves of different PVA−Sensors; (**f**) Equivalent resistance of different PVA−Sensor; (**g**) Charge transfer resistance of different PVA−Sensor; (**h**) Electrical conductivity of different PVA−Sensor; (**i**) Water content of different PVA−Sensor.

**Figure 4 sensors-24-06197-f004:**
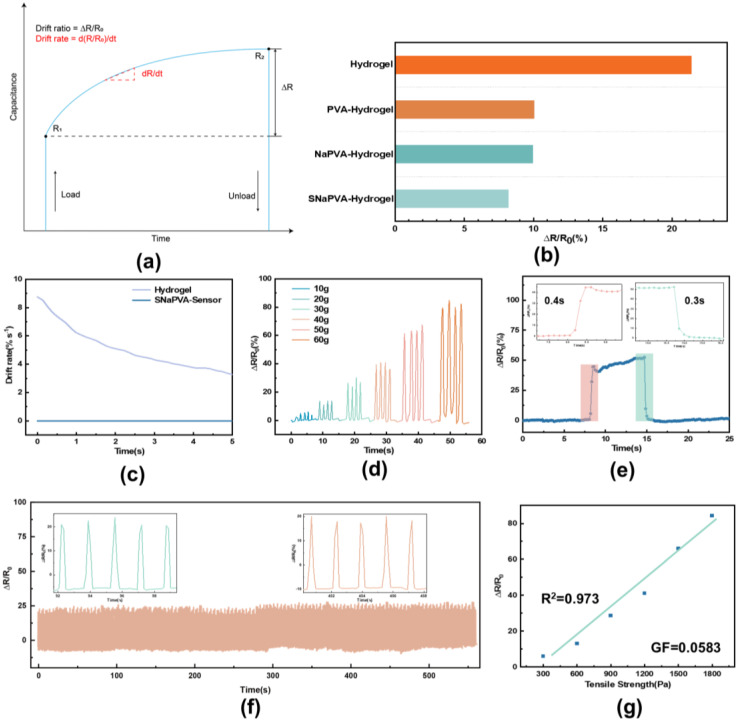
Sensing performance of SNaPVA−Sensor. (**a**) Definition of drift ratio and drift rate; (**b**) Drift ratio of conventional hydrogel sensor and different PVA−Sensor; (**c**) Drift ratios of conventional hydrogel sensors and SNaPVA−Sensor; (**d**) Current response versus time for SNaPVA−Sensor at different pressures; (**e**) Response and recovery time of SNaPVA−Sensor to pressure at 1 kPa. (**f**) Cyclic stability of the SNaPVA−Sensor for a long cycle loading and unloading cycle at 1.2 kPa. (**g**) Variation of ∆R/R_0_ of SNaPVA−Sensor with pressure.

**Figure 5 sensors-24-06197-f005:**
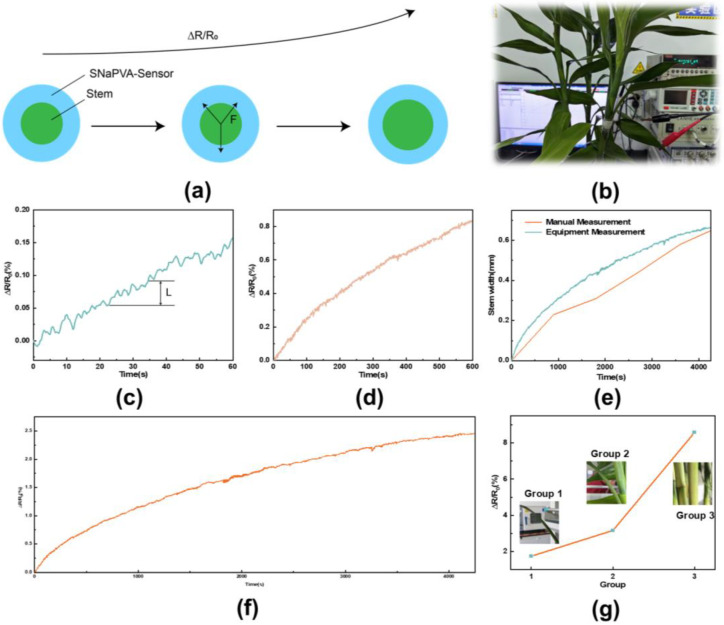
Performance of SNaPVA−Sensor for monitoring plant growth. (**a**) Model of SNaPVA−Sensor applied to plant stem; (**b**) Physical drawing of SNaPVA−Sensor monitoring plant stem growth; (**c**) SNaPVA−Sensor for short time plant growth monitoring; (**d**) SNaPVA−Sensor for long time plant growth monitoring; (**e**) Differences between SNaPVA−Sensor and manual monitoring; (**f**) SNaPVA−Sensor for long period plant growth monitoring; (**g**) SNaPVA−Sensor for different stem widths of the response.

**Figure 6 sensors-24-06197-f006:**
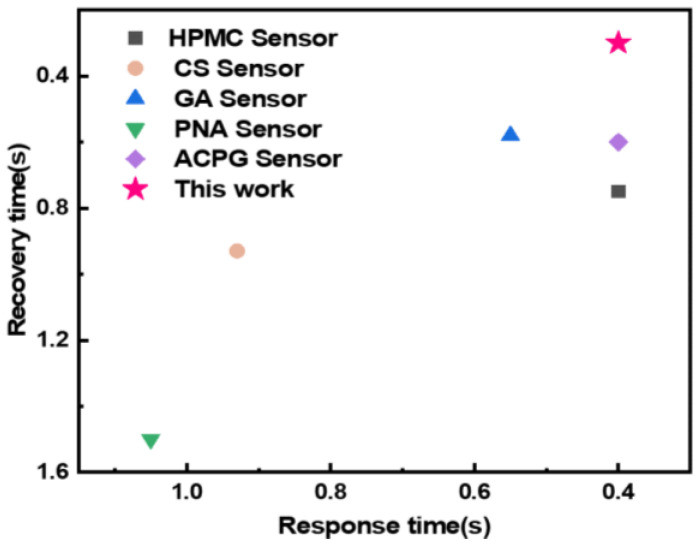
Comparison of response/recovery time of SNaPVA−Sensor and other sensor.

## Data Availability

The data that support the plots within this paper and other findings of this study are presented in the main article and the Appendix A. Additional data related to this paper may be requested from the corresponding authors upon reasonable request.

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
