# Peer review of "A Highly Sensitive, Low Creep Hydrogel Sensor for Plant Growth Monitoring"

_sensors, 2024, doi:10.3390/s24196197_

Round 1

Reviewer 1 Report

Comments and Suggestions for Authors

span lang="EN-US" The study presents the development of a highly sensitive, low-creep hydrogel sensor composed of polyvinyl alcohol (PVA) and acrylamide (AAm) for plant growth monitoring. The hydrogel exhibits a robust interpenetrating double network structure, enhanced by glycerol and cross-linking agents, which minimizes creep under external stress. Here are some comments:/span/p p class="MsoListParagraph" style="margin-left: 18.0pt; text-align: left; text-indent: -18.0pt; mso-char-indent-count: 0; mso-list: l0 level1 lfo1" align="left"1.     The authors should make a table and compare with other similar sensors in the literature.

2.     The authors should cite important recent literature about hydrogel sensor such as Doi:10.1155/2021/2225426; Doi:10.1007/s43939-024-00111-8

3.     The resolution of the figures is too low, please replace it.

4.     There should be a space between the number and the unit

Author Response

We thank the reviewers for their valuable comments. Please see the attachment for details.

Reviewer 2 Report

Comments and Suggestions for Authors

The authors developed a highly sensitive, low-creep hydrogel sensor composed of polyvinyl alcohol and acrylamide for plant growth monitoring. While the idea of using hydrogel strain sensors in the agricultural field is interesting, the choice of hydrogel material in this study lacks novelty. Moreover, the manuscript would benefit from further refinement in its overall structure and logic. The following points should be addressed to improve the manuscript before publication:

1. Why did the authors choose polyvinyl alcohol and acrylamide to prepare the hydrogel sensors? What are the specific advantages of these materials over other potential candidates?

2. The authors should elaborate on the mechanism by which the hydrogel sensor monitors plant growth. This aspect is currently underexplored in the manuscript.

3. On page 4, line 156, the authors state, “Sodium chloride as a dynamic ion enhances sensing performance.” The rationale behind this statement needs further explanation. It is recommended that the authors provide additional characterization or data to support this claim.

4. What was the purpose of testing the electrochemical properties of the hydrogels? The manuscript should clarify the relevance of this test in the context of the sensor's intended application.

5. Hydrogel strain sensors require not only effective sensing capabilities but also robust mechanical properties, especially for monitoring plant growth. The manuscript currently lacks research on the mechanical properties of the hydrogel sensors. The authors are advised to consult recent studies on hydrogel sensors (e.g., Adv. Mater. 2022, 34, 2107106; Adv. Funct. Mater., 2023, 33, 2305705…) and include relevant experiments to address this aspect.

6. How do the authors plan to experimentally validate pressure sensing by hydrogel sensors for plant growth monitoring? Is there an established standard for quantifying this?

7. In Figure 3f, the resistance change appears unstable. The authors should consider retesting this aspect to ensure the reliability of their data.

8. Both the sensitivity and linearity of a sensor are critical factors in evaluating its performance. However, the manuscript lacks relevant data on these parameters. The authors should include this information to provide a comprehensive evaluation of their sensor.

9. The authors should ensure consistency in the formatting of all cited references. This includes uniformity in the number of authors listed, journal names, publication years, volume numbers, and page number formats.

Comments on the Quality of English Language

The language still needs to be further improved and the format needs to be improved

Author Response

(The authors gave the same response as above.)

Reviewer 3 Report

Comments and Suggestions for Authors

The paper presents the development and performance of a stable and sensitive PVA hydrogel-based plant sensor. Addressing the creep response of hydrogel through the polymer design for flexible sensing is much needed. However, there is considerable work necessary for improving the write-up before publication. Introduction

·       Additional references from different places needed to support the statement, “Researchers have sought to integrate flexible sensors with plants to address the limitations of traditional rigid sensors[13-20].”

o   Yin, S. and Dong, L., 2024. Plant Tattoo Sensor Array for Leaf Relative Water Content, Surface Temperature, and Bioelectric Potential Monitoring. Advanced Materials Technologies, p.2302073.

o    Borode, T., Wang, D. and Prasad, A., 2023. Polyaniline-based sensor for real-time plant growth monitoring. Sensors and Actuators A: Physical355, p.114319.

·       Method Section

o   Add a schematic schematic figure showing the material process steps in Section 2.1

o   Need additional details for the process used

§ Line 103 “specific amounts of cross-linking agents and initiators were added to the mixture” Specify the amount.

§ Freeze thawing cycle: was it room temp thawing or under what condition since this process is deemed critical in the process

o   The line is stated early before any robustness and effectiveness is proven and so remove it from here. “The result is a robust and effective PVA-Sensor ready for various applications”

o   Renumber Characterized as Section 2.2 (not 2.3)

o   Provide details on the method of characterization. Multiple methods are listed (Ex, FT-IT. XRD, CV etc) but no details as what is being tested

§ Suggest expand to two sections with 2.2 Material characterization  and 2.3 Electrical response characterization and and provide details for each. For example, FTIR and XRD what peaks will indicate the material behavior, with any references to back that.

§ Same for other methods listed.

·       Results Section

o   Line 144-145: since the sensor is wound across, there will be movement from radial and longitudinal growth, depending on the plant type and age of plant. How does the sensor capture and differentiate between these two?

o   Line 148 to 160 is not a result. It should be placed in the methods when used to justify the choice of materials. Same for Figure 1b, which should be earlier.

o   Conclusions are drawn in the results section without a sound basis. Ex: Line 172-177

o   1c? shown in figure but not discussed in results below. What was the approach for the tensile test which is not described in the method.

o   Drift/sensitivity/ etc. is important, as shown in Figure 3, but the method does not discuss the setup used for those tests.

·       Overall, significant work is needed on results and methods. First, expand on the methods mentioned here, and then discuss the results in the context of the data. The paper as written jump a few steps and conclude without describing the method and/or presenting the results.

·       Discussion:

o   Why 315-316. Again first write what/why the enhancement before drawing a statement about the sensor

o   The discussion section needs to highlight the sensor performance in the context of sensors reported earlier. What are the advantages/disadvantages?

Comments on the Quality of English Language

No major english editing is needed

Author Response

(The authors gave the same response as above.)

Round 2

Reviewer 2 Report

Comments and Suggestions for Authors

The authors have addressed all my concerns. I recommend acceptance of this work.

Comments on the Quality of English Language

Language checking should be done through the manuscript.

Also, there are some formatting problems, like the cited references. Please correct them.

Author Response

(The authors gave the same response as above.)

Reviewer 3 Report

Comments and Suggestions for Authors

The authors have addressed most of the concerns in the response letter, but at few places missed adding the response/explanation in the  manuscript.

- In response to the freeze-thawing cycle,  explanation, and the two citations provided should be put in the paper in Section 2.4 of the method

- In response to previous lines 144-145, the response and explanation is not correct. The growth is both radial and longitudinal, and in fact is primarily longitudinal at the vegetative stages, and then radial in older stages, as explained in this work and others. Also not clear why "Today..."? So overall, provide the literature in the explanation on radial growth and this below on longitudinal growth, and then highlight what stages of plant growth this sensor is suitable and/or the limitations in measuring longitudinal growth or impact from that. 

Roy M, Mathew FM, Prasad A. Biomechanics of vascular plant as template for engineering design. Materialia. 2020 Aug 1;12:100747.

Author Response

(The authors gave the same response as above.)
